# Consideration of FedProx in Privacy Protection

Tianbo An, Leyu Ma [ID], Wei Wang *, Yunfan Yang, Jingrui Wang and Yueren Chen

School of Network Security, Changchun University, Changchun 130012, China; antb@mails.ccu.edu.cn (T.A.);
210701251@mails.ccu.edu.cn (L.M.); 210701242@mails.ccu.edu.cn (Y.Y.); 210701240@mails.ccu.edu.cn (J.W.);
210701245@mails.ccu.edu.cn (Y.C.)
* Correspondence: wangwei@mails.ccu.edu.cn

**Abstract:** As federated learning continues to increase in scale, the impact caused by device and data heterogeneity is becoming more severe. FedProx, as a comparison algorithm, is widely used as a solution to deal with system heterogeneity and statistical heterogeneity in several scenarios. However, there is no work that comprehensively investigates the enhancements that FedProx can bring to current secure federation algorithms in terms of privacy protection. In this paper, we combine differential privacy and personalized differential privacy with FedProx, propose the DP-Prox and PDP-Prox algorithms under different privacy budget settings and simulate the algorithms on multiple datasets. The experiments show that the proposed algorithms not only significantly improve the convergence of the privacy algorithms under different heterogeneity conditions, but also achieve similar or even better accuracy than the baseline algorithm.

**Keywords:** privacy protection; FedProx; differential privacy; personalized differential privacy

## 1. Introduction

The amount of application data uploaded, stored and used by various smart terminals is continuously increasing and federated learning (FL) [1], as a solution to achieve data sharing and fusion across devices and organizations is facing new challenges, such as privacy protection, system and data heterogeneity and the expansion of the scale of distributed devices.

Federated learning, represented by FedAvg, can deal with the problems of heterogeneity and high communication costs by allowing low user participation and local update optimization. In each iteration, FedAvg selects some user devices to participate in the computation and updates the model parameters by performing stochastic gradient descent (SGD) for a certain number of rounds, and a central server aggregates the local parameters of all users to update the global parameters. Since FedAvg is unable to perform a variable number of training rounds based on system constraints, devices that cannot complete a specified number of rounds within a specified time window are discarded in cases of system heterogeneity. To enhance the anti-heterogeneity of federated learning algorithms, Li et al. [2] proposed FedProx, which addresses system heterogeneity and statistical heterogeneity by allowing the occurrence of inadequately trained local models and adding proximal terms to the original loss function, respectively.

To achieve privacy protection for federated learning, secure federated learning can currently be realized via algorithms such as secure multi-party computation [3,4], homomorphic encryption [5,6] and differential privacy (DP) [7]. Compared to the first two approaches, DP can provide rigorous and robust privacy protection based on low computational and communication costs and thus is widely used in various modeling algorithms to protect privacy. The current work on applying DP for privacy protection can be divided into two categories based on the privacy settings: uniform privacy budget settings and personalized privacy budget settings [8]. Points of interest regarding the work in the uniform privacy budget scenario are reducing the impact of added noise on the model through

privacy amplification [9], user/data filtering and parameter selection [10–13]. This kind of work does not take the user's personalized privacy needs into consideration, resulting in a certain amount of wasted privacy; the focus of the work in the personalized privacy budget scenario is to realize the personalization of privacy protection by optimizing the user selection and privacy-budget-allocation mechanism [14–17].

The most representative algorithm for applying DP to the federated learning scenario is DP-SGD [18], which closely combines DP and SGD. By adding noise to the clipped gradient values, DP-SGD can effectively protect the data information, so DP-SGD is adopted as a local solver for model updating in most of the work. However, DP-SGD is proposed without specific consideration of heterogeneity.

Inspired by the DP-SGD algorithm and based on the advantages of FedProx in addressing system heterogeneity and statistical heterogeneity, this paper combines DP and PDP with FedProx and proposes two algorithms, DP-Prox and PDP-Prox, respectively, to explore the utility of privacy algorithms of FedProx under different privacy budget settings. The primary contributions of this study are as follows:

- We propose the DP-Prox algorithm under a unified privacy budget scenario to improve the convergence of the model algorithm under the condition of heterogeneity.
- We propose the PDP-Prox under a personalized privacy budget scenario to improve the balance between privacy and utility through multiple rounds of the adaptive sampling mechanism.
- We conduct a series of comparison experiments with the baseline algorithm on both synthetic and real datasets to demonstrate that our proposed algorithm is not only more adaptable to heterogeneous environments, but also achieves up to nearly 6% improvement in algorithmic accuracy when compared to the commonly used SGD algorithm.

This study follows the following format: In Section 2, we give a summary of the background information and associated research on FL and DP. We examine recent relevant studies in Section 3. In Section 4, we go into further detail about our proposed algorithm and the mechanisms that connect to it and in Section 5, we perform an experimental evaluation of the proposed method. Finally, Section 6 brings the paper to a close.

## 2. Preliminaries

In this section, we introduce three key elements of the new modeling algorithm, namely DP, PDP and the FL framework FedProx.

### 2.1. Differential Privacy and Personalized Differential Privacy

We start by introducing the idea of $(\varepsilon, \delta) - DP$ [19]. Due to the originally proposed $\varepsilon - DP$'s strict privacy protection [20], a sizable privacy budget is required in order to implement it in practical applications. The size of the privacy budget is inversely correlated with the practicality of the algorithm and excessive use of privacy counting will significantly lower the accuracy of the computation results. Since the method can still satisfy the DP within a certain gap, Dwork et al. [19] added a relaxation element $\varepsilon$ to the original specification. $(\varepsilon, \delta) - DP$ is consequently a relaxation of $\varepsilon - DP$.

**Definition 1.** *($(\varepsilon, \delta) - DP$) A randomized mechanism $\mathcal{M} : \mathcal{D} \to R$ with value domain $\mathcal{D}$ and range R satisfies $(\varepsilon, \delta) - DP$ if for any two adjacent inputs $d, d' \in \mathcal{D}$ such that $\| d - d' \| \leq 1$ and for any subset of outputs $O \subseteq R$ it holds that*

$$Pr[\mathcal{M}(d) \in O] \leq e^{\varepsilon} Pr[\mathcal{M}(d') \in O] + \delta \tag{1}$$

The DP process can be realized via random response (RR) [21], Laplace noise [22] or Gaussian noise [19]. Gaussian noise is chosen throughout this study since it is more adaptable in data processing and is ideal for developing $(\varepsilon, \delta) - DP$ mechanisms.

**Definition 2.** *(Gaussian Mechanism) For a function* $f : \mathcal{D} \rightarrow R$ *with sensitivity* $\Delta_2(f) = max_{d,d' \in \mathcal{D}} \parallel f(d) - f(d') \parallel_2$, *the random mechanism* $\mathcal{M}(d) = f(d) + \mathcal{N}(0, \sigma^2)$ *obeys* $(\varepsilon, \delta) - DP$ *if for any* $\delta \in (0,1)$, *given random noise obeying a normal distribution* $\mathcal{N}(0, \sigma^2)$, *where:*

$$\varepsilon \geq \frac{\sqrt{2ln(1.25/\delta)}}{\frac{\sigma}{\Delta_2 f}} \tag{2}$$

As can be seen from Definition 1, the traditional DP ignores the various privacy demands and preferences of users and sets the privacy criteria of all users to a standard privacy budget value of $\varepsilon$; meanwhile, setting a uniform privacy budget to satisfy the privacy demands of all users will negatively affect the utility of the modeling algorithm. In order to strike a compromise between privacy and utility, Jorgensen et al. [8] presented PDP ($\varepsilon - PDP$), which improves the utility of the algorithm by adopting various privacy budget values. Based on $\varepsilon - PDP$, Heo et al. [23] expanded it to $(\phi, \Delta) - PDP$.

**Definition 3.** *($(\phi, \Delta) - PDP$) In the context of the privacy budget* $\phi$ *and the set of users* $U$, *a randomized mechanism* $\mathcal{M} : \mathcal{D} \rightarrow R$ *satisfies* $(\phi, \Delta) - PDP$, *if for any two adjacent datasets* $D, D' \subset \mathcal{D}$, $D \xrightarrow{d} D'$ *and for any subset of outputs* $O \subseteq R$, *it holds that*

$$Pr[\mathcal{M} \in O] \leq e^{\varepsilon_i} \times Pr[\mathcal{M}(D') \in O] + \delta_i \tag{3}$$

*where* $u_i \in U$ *is the user associated with tuple d,* $\varepsilon_i \in \phi$ *represents the privacy needs of user i and* $\delta_i \in \phi$ *represents the probability of user i's information leakage.*

Definition 3 is a generalization of Definition 1 and $\phi - PDP$. A definition transforms to $\phi - PDP$ if for all users $u \in U$ the value of $\delta_i$ is 0. The definition transforms to $(\varepsilon, \delta) - DP$ if for all users $u \in U$, $\varepsilon_i = \varepsilon$ and $\delta_i = \delta$.

**Sampling Mechanism**

The sampling mechanism, which is based on the sampling approach of privacy amplification and may effectively limit the loss of privacy, was initially put out by Jorgensen et al. [8] and can be employed in any $\varepsilon - DP$ algorithms. Poisson distribution sampling, uniform sampling and shuffle shuffling [24] are the sampling techniques that are employed most often. Heo et al. [23] proposed a sampling technique that fulfills $(\varepsilon, \delta) - DP$ by extending the mechanism described in [8]. This also established that the mechanism $\mathcal{M}_s$ satisfies $(\phi, \Delta) - PDP$ when $\Delta = \{\delta_i | \delta_i = \pi_i \delta, u_i \in U\}$.

**Definition 4.** *($(\varepsilon, \delta) - DP$ Mechanism with Sampling) We use $SM_{PDP}$ to represent a randomized algorithm that satisfies $(\tau, \delta) - DP$. For a dataset $D \subset \mathcal{D}$, a privacy budget $\phi = \{\varepsilon_1, \dots, \varepsilon_n\}$ and a sampling threshold $\tau \in [min\phi, max\phi]$, we let $RS(D, \phi, \tau)$ represent the preprocessing step of probabilistically independent sampling of each data point $x \in D$:*

$$\pi_i = \begin{cases} \frac{e^{\varepsilon_i} - 1}{e^{\tau} - 1} & \text{if} \quad \varepsilon_i < \tau \\ 1 & \text{otherwise} \end{cases} \tag{4}$$

The output of a sampling mechanism $\mathcal{M}_s$ can be defined as

$$\mathcal{M}_s(D, \phi, \tau) = SM_{PDP}(RS(D, \phi, \tau)) \tag{5}$$

when $\Delta = \{\delta_i | \delta_i = \pi_i \delta, u_i \in U\}$, the mechanism $\mathcal{M}$ satisfies $(\phi, \Delta) - PDP$.

*2.2. FedProx*

FL securely enables decentralized data sharing by coordinating data collection, training and fusion across several dispersed end devices and a central server. Let $k$ represent the set of all terminals of size $| K |= N$ and $\mathcal{D}_k$ denote all possible data distributions. Let $f_k(w; x_k)$ stand for the loss function of terminal $k$ over model $w$ and sample $x_k$ and $F_k(w) :=$

$\mathbb{E}_{x_k \sim \mathcal{D}_k}[f_k(w; x_k)]$ represent the loss function (possibly non-convex) of terminal $k$. The FL method minimizes the objective function by solving

$$\min_w \left\{ f(w) = \sum_{k \in K} \frac{n_k}{n} F_k(w) \right\} \tag{6}$$

to perform collaborative training, where $n = \sum_{k \in K} n_k$ is the total value of the size of all terminal datasets.

Each iteration of FL uses a different sampling technique for device selection in an effort to lower communication costs. Utilizing local solvers, the chosen devices optimize each of their local objective functions before uploading the modified local model parameters to the centralized server. The central server changes the global model parameters by combining the provided specific update parameters. Most existing methods only permit a certain number of uniform training cycles for each participating device, not taking system heterogeneity into account. Since various devices have varied computing, storing and communication capacities, the completion time of equivalent training rounds will be sequential. A frequent practice is to discard the device models that do not successfully complete the training within the allotted time because waiting for all devices to finish can slow down the training process of the entire system. The total training accuracy might however be impacted by model bias caused by the abandoned devices.

Flexibly adapting the number of training cycles for each device in each round can be realized by solving the inexact solution of each local objective function in FL.

**Definition 5.** *($\gamma - $ inexact solution) For a function* $h(w; w_0) = F(w) + \frac{\mu}{2}\|w - w_0\|^2$, $\gamma \in [0, 1]$, *we say that* $w^*$ *is an* $\gamma - $ *inexact solution of* $\min_w h(w; w_0)$ *if* $\|\nabla h(w^*; w_0)\| \leq \gamma \|\nabla h(w; w_0)\|$, *where* $\nabla h(w; w_0) = \nabla F(w) + \mu(w - w_0)$. *Notice that smaller values of* $\gamma$ *correspond to higher accuracy.*

Li et al. [2] extend Definition 3 by defining $\gamma_k^t$ for each device in each round of iterations, i.e., FedProx allows each device to solve its own local objective function inexactly according to its own situation and is not configured with a uniform value of $\gamma$ across devices and the variable number of local iterations can be viewed as a proxy for $\gamma_k^t$.

**Definition 6.** *($\gamma_k^t - $ inexact solution [2]) For a function* $h_k(w; w_t) = F_k(w) + \frac{\mu}{2}\|w - w_t\|^2$, $\gamma \in [0, 1]$, *we say that* $w^*$ *is an optimal solution of* $\min_w h_k(w; w_t)$ *if* $\|\nabla h_k(w^*; w_t)\| \leq \gamma_k^t \|\nabla h_k(w_t; w_t)\|$, *where* $\nabla h_k(w; w_t) = \nabla F_k(w) + \mu(w - w_t)$. *Notice that smaller values of* $\gamma$ *correspond to higher accuracy.*

Another manifestation of heterogeneity is statistical heterogeneity, i.e., the data are non-independently and identically distributed (non-IID). After mitigating the effects of system heterogeneity, more devices will typically take part in each round of iterative training and the more devices that participate locally in model updating, the greater the divergence caused by statistical heterogeneity in the system is likely to be. Therefore, FedProx adds a proximal term to limit the deviation of local updates between rounds while solving the local function. Specifically, instead of only updating the model parameters via minimization of the local function $F_k(w)$, device k employs its local selection solver to approximate and minimize the following target $h_k$:

$$\min_w \left\{ h_k(w; w^t) = F_k(w) + \frac{\mu}{2}\|w - w^t\|^2 \right\} \tag{7}$$

FedProx is modified from FedAvg in terms of the inexact solution and proximal term and transforms to FedAvg when (1) the proximal term is not invoked, i.e., $\mu = 0$, (2) system heterogeneity is not taken into account, i.e., $\gamma$ is a constant, and (3) the local solver is chosen to be SGD. Because it can handle the negative effects of system heterogeneity and statistical

heterogeneity on the algorithm, FedProx is more generic in real settings than FedAvg and is therefore more adaptable.

## 3. Related Work

The current work on privacy-protection research through DP in FL can be classified into centralized DP (GDP) [25,26] and localized DP (LDP) [10,11,27–31] based on the class of DP. Liu et al. [25] introduced DP into FL for the first time to achieve user-level privacy protection by protecting the user's entire dataset; Lian et al. [26] further assumed that the communication channel is not entirely secure and proposed an NbAFL scheme that satisfies the DP requirements by adding appropriate noise perturbations to the client and server. In contrast, LDP achieves data privacy locally to the user based on the assumption of unreliable third-party work, which can significantly enhance the privacy protection. Hu et al. [27] proposed LDP-Fed, an FL system with LDP guarantees. Given that different dimensions of data have varying degrees of importance, Liu et al. [10] proposed a fedSel algorithm to reduce the amount of noise injection by choosing the Top-k dimensions based on the contribution of the dimensions in the SGD iteration. In order to reduce the communication cost, Lian et al. [11] did not use common methods such as random selection through the client, but designed a layer-based parameter-selection method to select valuable parameters for global aggregation, while Geyer et al. [29] provided DP guarantees by adding noise to locally uploaded parameters in a personalized FL framework.

Even though the literature on privacy preservation and trade-off optimization is extensive, all known efforts operate on a single privacy budget framework. The algorithm's usefulness is significantly impacted by protecting data from diverse users with the same degree of security and the uniform protection strategy does not satisfy the growing need for personalized privacy protection.

PDP [8,32,33], on the other hand, delivers different levels of privacy protection for users with various privacy demands by setting different privacy budgets on top of the uniform privacy protection level of DP. Current PDP research under FL focuses on personalized LDP (PLDP) [14–16,34–36]. Facing the two major challenges of user privacy level selection and model optimization, Shen et al. [14] proposed the PLU-FedOA algorithm based on the PVLDP [11] to solve it in modules. Based on the stochastic response mechanism, Chen et al. [34] proposed a perturbation algorithm, PDPM, to satisfy the PLDP. Shen et al. [36] designed three models based on the existing LDP mean estimation scheme in order to provide customized privacy for each user. All of the above works incorporate the concept of PDP into FL in different perspectives, but they do not take into account the wastage of the user's personalized privacy budget due to user selection and noise addition in the process of multiple iterations. The most recent study [23] was based on Ada-PDP, as proposed by Niu et al. [32]. It extends the DP-SGD algorithm to PDP-SGD through a utility-sampling mechanism and recycles the privacy budget wasted in each iteration.

All of the above works aim at optimizing the algorithm to achieve a higher level of privacy protection and do not view the impact of heterogeneity on the model as a major research issue. In this regard, in this study, we use FedProx as the base framework and introduce the concepts of DP and PDP, respectively, to propose new algorithms to evaluate the performance of FedProx in terms of privacy protection in different heterogeneous scenarios.

## 4. Differential Privacy in FedProx

As one of the most popular deep learning algorithms under the DP mechanism, DP-SGD trains the model parameters by minimizing the empirical loss function. The model is trained in a total of T rounds. In each round of training, firstly, a subset of the whole training set is selected by subsampling with probabilistic no-putback sampling; in the second step, the gradient of the loss function is computed for each subset of samples; in the third step, the gradient is clipped by the gradient paradigm thresholding; in the fourth step, Gaussian noise is added to the clipped gradient and all the local parameters are aggregated;

and in the fifth step, the model parameters are updated by the gradient that satisfies the DP. The specific algorithm flow is shown in Algorithm 1.

---

**Algorithm 1:** Differential Privacy SGD (DP-SGD).

1 **Input:** Example $D = \{x_1, \ldots, x_n\}$, loss function $\mathcal{L}(\theta) = \frac{1}{N}\sum_i \mathcal{L}(\theta, x_i)$, learning rate $\eta_t$, noise scale $\sigma$, group size L. **Initialize:** $\theta_0$ **for** $t \in [T]$ **do**

2 $\quad$ $L_t = qL, q = L/N$ ; $\hfill$ /* subsampling */

3 $\quad$ $g_t(x_i) \leftarrow \Delta_{\theta_t}\mathcal{L}(\theta_i, x_i), i \in L_t$ ; $\hfill$ /* compute gradient */

4 $\quad$ $\bar{g}_t(x_i) \leftarrow g_t(x_i)/\max(1, \frac{\|g_t(x_i)\|_2}{C})$ ; $\hfill$ /* clip gradient */

5 $\quad$ $\tilde{g}_t \leftarrow \frac{1}{L}(\sum_i \bar{g}_t(x_i) + \mathcal{N}(0, \sigma^2 C^2 I))$ ; $\hfill$ /* add noise and aggression */

6 $\quad$ $\theta_{t+1} \leftarrow \theta_t - \eta_t \tilde{g}_t$ ; $\hfill$ /* take gradient step */

7 **end**

8 **Output:** $\theta_T$

---

### 4.1. DP-Prox

In this study, we introduce DP into the FedProx framework and modify the DP-SGD algorithm in two ways. On the one hand, the objective function for calculating the gradient adds the proximal term $\gamma$ to address the statistical heterogeneity and the bias introduced by the addition of noise; on the other hand, the imprecise solution is computed for the model and the impact of the system heterogeneity on the model is addressed by setting an unfixed value for the subset of devices participating in the training. The specific algorithm flow is described in Algorithm 2.

---

**Algorithm 2:** Differential Privacy FedProx (DP-Prox) (Proposed Framework).

1 **Input:** inexact parameter $\mu, \gamma$, devices $k = \{k_1, \ldots, k_n\}$, subset size $K_R$, number of rounds R, sample probability $q_k$ **procedure** Server Execution **Initialize:** $w^0$ **for** *each round* $r = 1, \ldots, R$ **do**

2 $\quad$ $S_r \leftarrow q_k \times \{k_1, \ldots, k_n\}$ ; $\hfill$ /* subsampling */

3 $\quad$ **for** *each devices* $k \in S_r$ **do**

4 $\quad\quad$ $\Delta w^{t+1} \leftarrow LocalUpdate(k, w^t)$; $\hfill$ /* client update */

5 $\quad$ **end**

6 $\quad$ $w^{r+1} \leftarrow w^r + \frac{1}{K}(\sum_{k=1}^{K} \Delta w_k^{r+1}/\max(1, \frac{\|\Delta w^{r+1}\|^2}{C}) + \mathcal{N}(0, \sigma^2 C^2))$ ; $\hfill$ /* global aggression */

7 **end**

8 **function** LocalUpdate $(k, w^r)$ **for** *each local epoch* $t = 1, \ldots, T$ **do**

9 $\quad$ $w^{r+1} \approx \arg\min h(w; w^r) = F(w) + \frac{\mu}{2}\|w - w^r\|^2$; $\hfill$ /* update parameter */

10 **end**

11 $\Delta w^{r+1} = w^{r+1} - w^r$ return $\Delta w^{t+1}$

---

### 4.2. PDP-Prox

We propose the PDP-Prox algorithm based on DP-Prox. By introducing PDP into FedProx using a personalized sampling mechanism, the PDP-Prox algorithm can set distinct privacy budgets on different user devices according to the user's personalized privacy needs under the FedProx framework to reduce the waste of the privacy budget.

**Multi-round adaptive sampling mechanism**

Since a uniform value is used in DP-Prox, it is challenging to make a trade-off between privacy protection and model utility. As a result, Nui et al. [32] proposed a utility-aware sampling mechanism to implement PDP, which improves the utility of the algorithm while accommodating users' various privacy needs.

The realization of PDP is mainly achieved through a two-step operation of sampling and noise addition, while the existence of randomness in both will cause sampling error and noise error, respectively. Sampling error refers to the impact of the sampling operation on the training effect of the original dataset and a high percentage of sampling corresponds to a low sampling error, whereas noise error refers to the degree of perturbation of the training results by different degrees of privacy requirements, with a high privacy requirement corresponding to a high noise error. The sampling mechanism mentioned in Definition 4 has some limitations. By defining the sampling threshold, the original dataset is divided into the unsampled part and the sampled part, and the users in the unsampled part will cause a sampling error due to the wastage of the privacy budget as the data are not used; meanwhile, if the privacy budget of the users in the sampled part is higher than the threshold, the users' data will be protected by the noise, which is higher than the privacy requirement, causing a noise error. Sampling error and noise error are defined, respectively, as

$$
\begin{aligned}
\omega_s(\phi, \tau) &= \sum_{i:\varepsilon_i < \tau, \varepsilon_i \in \phi} \varepsilon_i (1 - \pi_i) \\
\omega_n(\phi, \tau) &= \sum_{i:\varepsilon_i > \tau, \varepsilon_i \in \phi} \varepsilon_i - \tau
\end{aligned}
\tag{8}
$$

Therefore, the selection of the sampling threshold needs to measure both types of errors to lessen the influence of the errors on the algorithm. Since the adjustment of the sampling threshold has an opposite effect on the sampling error and noise error, a large threshold reduces the impact of the noise error on the algorithm; however, a drop in the sampling probability results in an increase in sampling error. To achieve the best sampling threshold in each computation round, we determine the minimum value of the utility loss function and allow the weights of the two types of errors be automatically modified according to the percentage of weights via adaptive means. The utility loss function can be defined as

$$
waste(\phi, \tau) = \frac{\omega_s}{\omega_s + \omega_n} \times \sum_{i:\varepsilon_i < \tau, \varepsilon_i \in \phi} \varepsilon_i (1 - \pi_i) + \frac{\omega_n}{\omega_s + \omega_n} \times \sum_{i:\varepsilon_i > \tau, \varepsilon_i \in \phi} (\varepsilon_i - \tau)
\tag{9}
$$

For a given $\phi$, the optimal $\tau$ can be obtained by solving the following optimization problem

$$
\begin{aligned}
&\min \quad waste(\phi, \tau) \\
&\text{s.t.} \quad \min \phi \leq \tau \leq \max \phi
\end{aligned}
\tag{10}
$$

The user data are sampled using the optimal sampling threshold calculated in each round and the privacy budget remaining after each round of sampling is utilized in multiple iterations until the remaining value of the privacy budget is less than a given value.

**Algorithm Process**

We now apply the multi-round adaptive sampling mechanism to DP-Prox in order to implement PDP-Prox. We define $R$ as the overall iteration round count, $n$ as the number of iterations in each round and $\epsilon_R$ as the target round $R$ privacy budget. Each round's noise scale is calculated depending on the sampling threshold, which is initially calculated by the utility loss function. Then, using the sampling approach described in Definition 4, we select the user devices that will participate in the calculation for each round and we use the noise scale computation function described in [23] to calculate the variance parameter of the Gaussian noise for each round. The model parameters are then iteratively computed using DP-Prox and the privacy accountant is computed using RDP accountant [37]. Finally, each device updates its remaining privacy budget. Figure 1 schematizes the algorithm flow, while Algorithm 3 describes the particular algorithm flow.

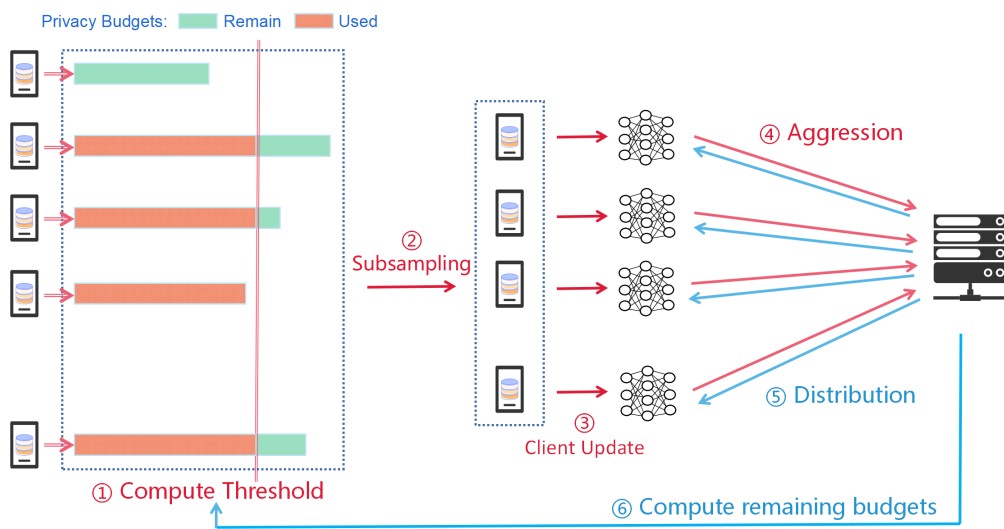

**Figure 1.** PDP-Prox algorithm flow diagram.

---

**Algorithm 3:** Personalized DP-Prox (PDP-Prox) (Proposed Framework).

---

1 **Input:** inexact parameter $\mu, \gamma$, devices $k = \{k_1, \ldots, k_n\}$, subset size $K_R$, privacy budgets $\phi = \varepsilon_i, \ldots \varepsilon_n$, number of rounds $R$, probability of privacy leakage $\delta$
   **procedure** Server Execution **Initialize:** $w^0, \tau$ **for** *each round $r = 1, \ldots, R$* **do**

2    $\tau_R \leftarrow \min(waste(\phi, \tau), \min \phi \leq \tau \leq \max \phi)$ ;      /* compute threshold */

3    $\sigma \leftarrow NoiseMultiplier(\tau_R, \delta, R)$;      /* compute noise multiplier */

4    $S_r \leftarrow M_s(K, \phi, \tau_R)$;      /* subsampling */

5    **for** *each devices $k \in S_r$* **do**

6      $\widetilde{\Delta w}_k^{r+1} \leftarrow LocalUpdate(k, w^t)$;      /* client update */

7    **end**

8    $w^{r+1} \leftarrow w^r + \frac{1}{K_r} \sum_{k=1}^{K_r} \widetilde{\Delta w}_k^{r+1}$ ;      /* global aggression */

9    $\varepsilon' \leftarrow Accountant(\sigma, k, R)$;      /* compute remain budgets */

10    $\varepsilon_k \leftarrow \max(\varepsilon_k - \varepsilon'), \forall i \in \{i | k_i \in S_r\}$

11 **end**

12 **function** LocalUpdate $(k, w^r)$ **for** *each local epoch $t = 1, \ldots, T$* **do**

13    $w^{r+1} \approx \arg \min h(w; w^r) = F(w) + \frac{\mu}{2} \|w - w^r\|^2$;      /* update parameter */

14 **end**

15 $\Delta w^{r+1} = w^{r+1} - w^r$   $\widetilde{\Delta w}_k^{r+1} = \Delta w_k^{r+1} / \max(1, \frac{\|\Delta w^{r+1}\|^2}{C}) + \mathcal{N}(0, \sigma^2 C^2)$ ;      /* add noise */

16 return $\widetilde{\Delta w}_k^{r+1}$

---

## 5. Experiment

In this section, we compare our proposed approach with the respective baseline models under unified privacy budget (DP) and personalized privacy budget (PDP) settings, respectively, to assess the usefulness of the FedProx-based privacy model and the extent to which each parameter affects the model.

### 5.1. Setting

Datasets. We use one synthetic dataset and two real datasets. Synthetic dataset [2]: for each device k, the samples $(X_k, Y_k)$ are generated according to the model $y = argmax$ $(softmax(Wx + b))$, $x \in b_k \sim \mathcal{N}(u_k, 1)$, $W \in R^{10 \times 60}$, $b \in R^{10}$. We follow the model $W_k \sim \mathcal{N}(u_k, 1)$, $b_k \sim \mathcal{N}(u_k, 1)$, $u_k \sim \mathcal{N}(0, \alpha)$; $x_k \sim \mathcal{N}(v_k, \sum)$, where the covariance matrix

$\Sigma$ is diagonal with $\Sigma_{i,j} = j^{-1.2}$. Each element in the mean vector $v_k$ is drawn from $\mathcal{N}(B_k, 1)$, $B_k \sim \mathcal{N}(0, \beta)$. Real datasets: 1. MNIST [38], which consists of handwritten images of numbers 0-9 and contains a total of 60,000 training samples and 10,000 test samples; and 2. Fashion-MNIST [39], which consists of images of 10 types of clothes and contains a total of 60,000 training samples and 10,000 test samples.

Baseline. Under a consistent privacy budget setting, the DPSGD algorithm acts as a baseline for comparison with the proposed DP-Prox method. In this experiment, our main goal is to determine if the proposed DP-Prox enables more robust convergence in the presence of many heterogeneous circumstances. Under a personalized privacy budget, the PDP-Prox algorithm is compared to the PDP-SGD method as a baseline. In this experiment, our main objective is to evaluate the extent to which the proposed PDP-Prox enhances the algorithm's accuracy compared to the baseline under several rounds of the adaptive sampling mechanism.

Environment. We use a simple convolutional neural network (CNN) in our experiments, which is modeled by two convolutional layers, two fully connected layers and two pooling layers. All experiments were conducted in a GPU: NVIDIA RTX 3080, CPU: 12 vCPU Intel(R) Xeon(R) Platinum 8255C CPU @ 2.50GHz environment. Due to the randomness of the algorithmic mechanism, all experiments were repeated five times with different random seeds.

### 5.2. DP-SGD and DP-Prox

We evaluate the effectiveness of FedProx in improving the convergence of DP algorithms through both system heterogeneity and statistical heterogeneity. In the experiments, the parameter $\delta = 1 \times 10^{-5}$ is chosen and the gradient clipping threshold C is set to the median value of the update parameters supplied by various individuals. In DP-Prox, we choose $\mu = 1$ as the default value of the parameter $\mu$ and a random imprecision value $\gamma_k^t$ is set for each device after each round of iteration.

**System heterogeneity**

We adjusted the system heterogeneity to 0%, 50% and 90% under a uniform privacy budget (DP) condition in order to assess the effect of system heterogeneity on the model, which corresponds to the scenarios of low system heterogeneity, medium system heterogeneity and high system heterogeneity, respectively (we model the number of devices that fail to complete training and exit the training process within a given communication round in a systematic heterogeneous environment by setting different heterogeneity ratios. The training results of such devices will not be used in the experiments). With a total of 100 rounds of iterations, there will be 10 user devices in each round.

Figure 2 illustrates how system heterogeneity affects the convergence of the model method to varying degrees: the higher the degree of heterogeneity, the poorer the convergence. Under various heterogeneous dissimilarity circumstances, the suggested DP-Prox's convergence robustness is superior to the baseline, while the DP-SGD method exhibits significant fluctuations. In terms of accuracy, the final performance of DP-Prox can also be slightly better than the baseline.

**Statistical heterogeneity**

We set $(\alpha, \beta) = (0,0), (0.5, 0.5), (1,1)$, respectively, to generate three non-identical distributed datasets. According to how the sample data are generated, $\alpha$ controls how much local models differ from each other and $\beta$ controls how much the local data at each device differ from those of other devices. In this experiment, we do not consider system heterogeneity (setting the parameter of system heterogeneity to 0%).

As shown in Figure 3, as data become progressively more heterogeneous, the convergence of the SGD-based privacy algorithm DP-SGD becomes worse. In terms of accuracy, the overall gap between the two types of algorithms is small, but as the number of training rounds increases, the DP-Prox algorithm shows a more pronounced advantage.

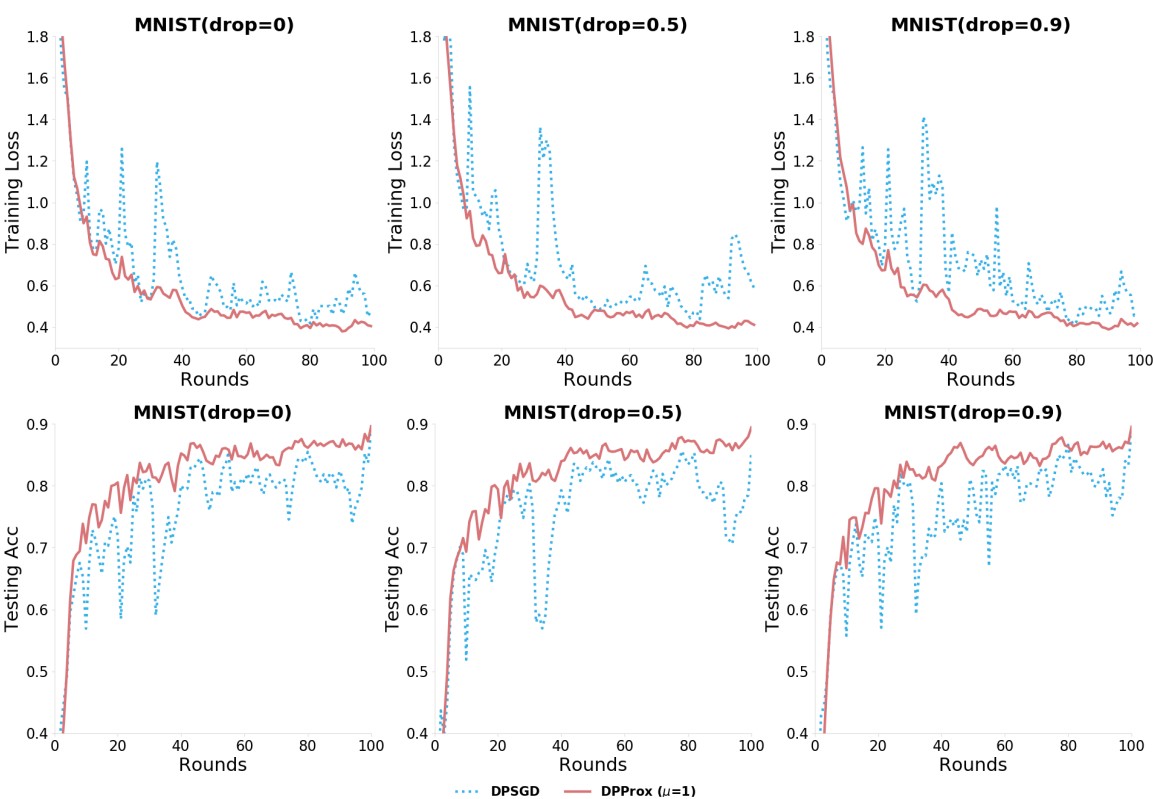

**Figure 2.** Training loss and test accuracy of the models trained on baseline (DP-SGD) and DP-Prox on the two datasets with the system heterogeneity setting.

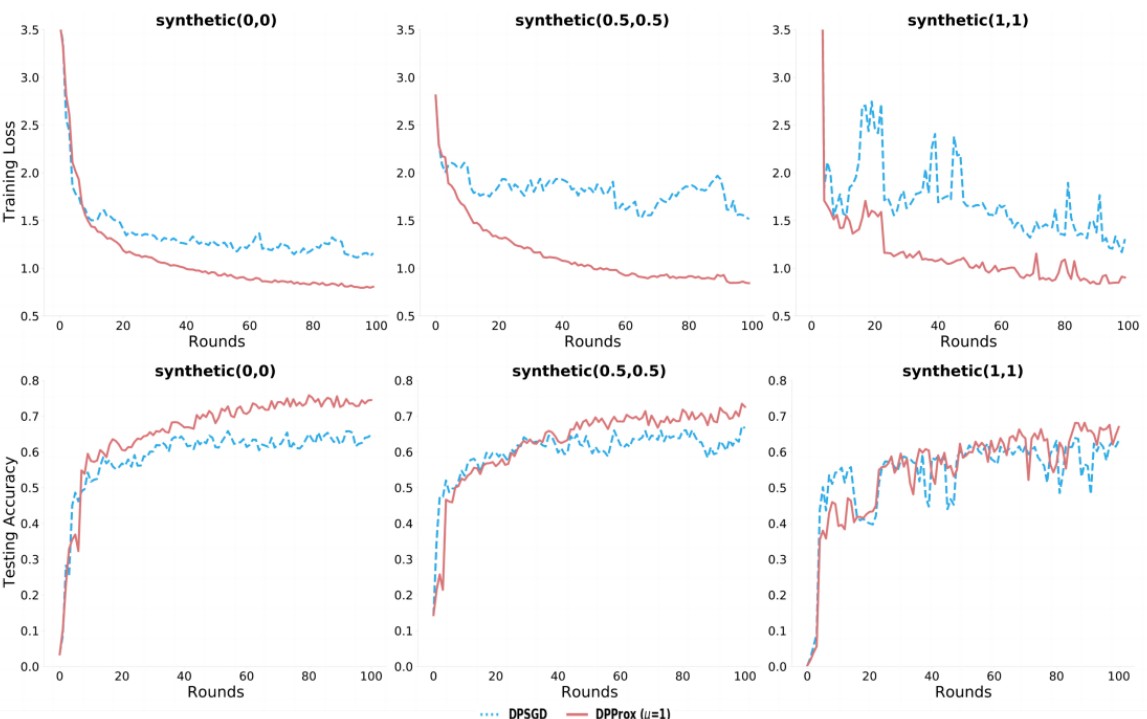

**Figure 3.** Training loss and test accuracy of the models trained on baseline (DP-SGD) and DP-Prox on the two datasets with the statistical heterogeneity setting.

**Parameter** $\mu$

For FedProx, the parameter $\mu$ is one of the most critical parameters. It regulates the disparity between the parameters and significantly affects the convergence of the algorithm. We set five different $\mu$ values for DP-Prox (0.01, 0.1, 0, 1 and 1.5, respectively) and set the total number of iteration rounds to 200 to evaluate the impact of the value of $\mu$ on the algorithm in the context of system heterogeneity, as shown in Figure 4. Based on the experimental findings, it can be deduced that the method converges poorly when the $\mu$ value is 0, i.e., when no dissimilarity measure is used in the derivation of the model parameters. This has a major detrimental effect on the algorithm. Selecting a smaller value of $\mu$ is likewise ineffective since the convergence is poorer when the $\mu$ value is lower. It is difficult for the model to calculate better values in the iterative process since choosing a smaller $\mu$ will make the parameters generated across various rounds extremely similar. This makes it simple to cause overfitting in the training, which has a higher influence on the algorithm's accuracy. The algorithm's convergence is substantially enhanced, the pace of convergence is quick and the fluctuation between rounds is minimal when the $\mu$ value is larger than 0. Since $\mu = 1$ performs marginally better than $\mu = 1.5$ in terms of accuracy, we may draw the conclusion that permitting the new model parameters to diverge greatly from the original parameters, i.e., choosing bigger amounts of dissimilarity, can also have a detrimental impact on the algorithm's utility.

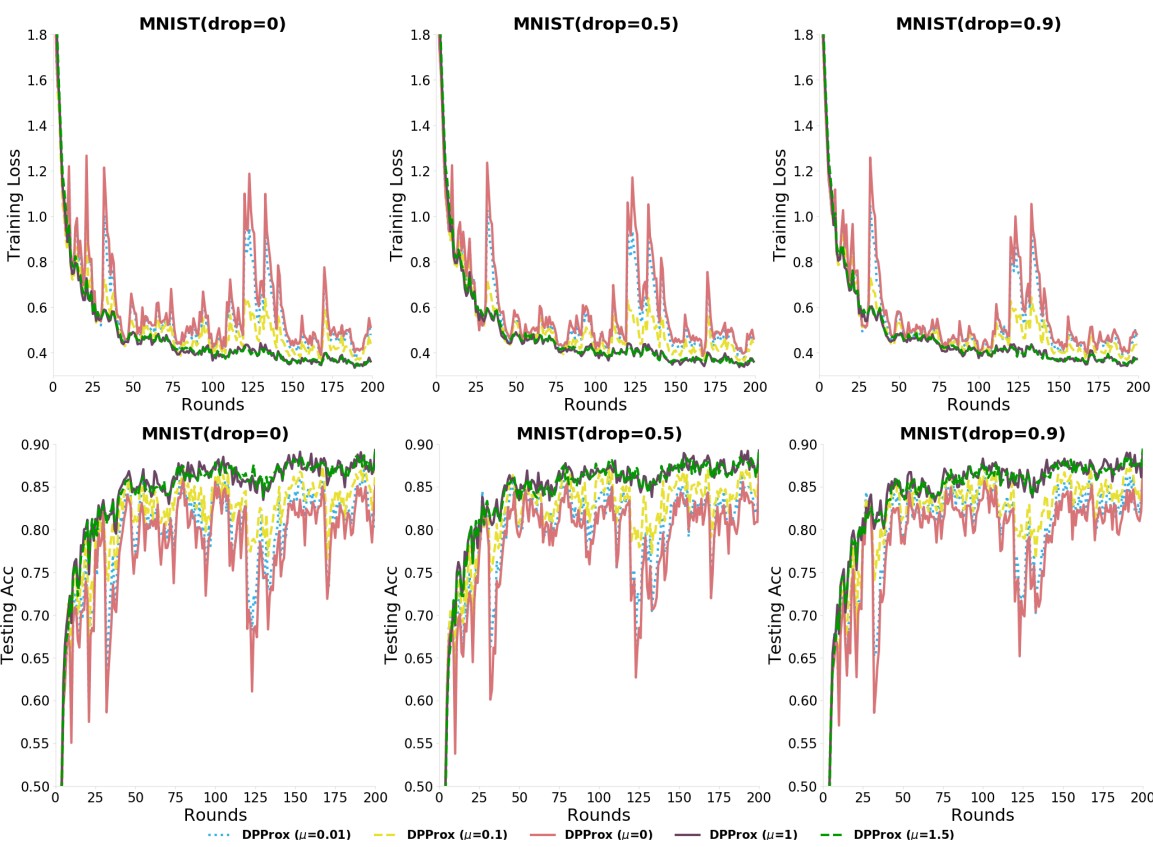

**Figure 4.** Impact of parameters on the loss and accuracy of the model.

### 5.3. PDP-SGD and PDP-Prox

PDPs can improve the utility of model algorithms while providing personalized protection by conserving privacy budgets. Therefore, in this section of experiments, we focus on evaluating the performance of the proposed PDP-Prox algorithm in terms of model utility.

Under the personalized privacy budget (PDP) setting, we use the PDP-SGD algorithm as a baseline and evaluate the proposed approach using the MNIST and Fashion-MNIST

datasets. Based on the experimental settings in [23], we set the privacy budget interval to $\varepsilon \in [0.5, 1]$ and apply an exponential distribution technique to create the privacy budget values for various users. Specifically, we split the privacy budget intervals into 20 groups on average and the number of users in different intervals is generated by $User(\varepsilon) = (e^{z\varepsilon} / \sum_{\varepsilon \in [0.5,1]} e^{z\varepsilon}) \times D_{train}$, where $D_{train}$ stands for the dataset size and $\varepsilon$ represents the privacy size of the group. The parameter z takes three values of $-0.2$, 0 and 0.2, which represent the three cases of users with generally larger privacy needs, users with the same privacy needs and users with generally smaller privacy needs, respectively (see Figure 5). The privacy budgets of the data points in the remaining unallocated privacy groups are set to be the maximum value of the privacy budget intervals as a consequence of rounding the exponential mechanism's findings.

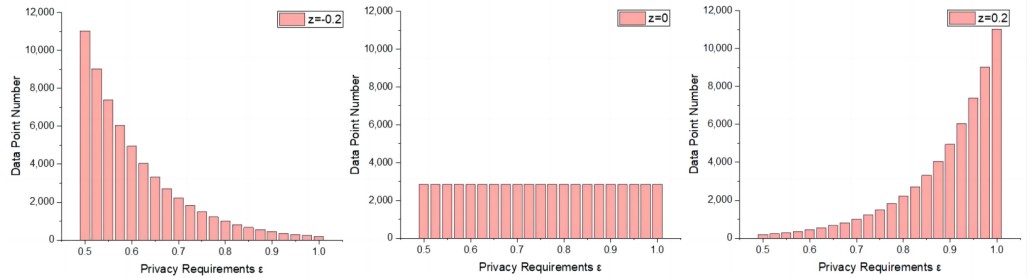

**Figure 5.** Schematic of privacy budget allocation.

Table 1 shows that the proposed PDP-Prox can achieve similar accuracy to the baseline on both the MNIST and Fashion-MNIST datasets and that the results of multiple experiments can be better than the baseline results. In scenarios with high privacy requirements, the accuracy can be increased by up to nearly 5%. Through multiple experiments, the PDP-Prox algorithm is verified to converge better than PDP-SGD and the mean function loss value of each experiment can be between 0.2 and 0.5 less than that of the PDP-SGD algorithm.

**Table 1.** Test accuracy and loss of the models trained on baseline (PDP-SGD) and PDP-Prox on the two datasets.

| Dataset | Skew | PDP-SGD | | PDP-Prox | |
|---|---|---|---|---|---|
| | | **Accuracy** | **Loss** | **Accuracy** | **Loss** |
| MNIST | $z = -0.2$ | $94.08 \pm 0.64$ | $0.68 \pm 0.07$ | $\mathbf{94.99 \pm 0.14}$ | $\mathbf{0.22 \pm 0.02}$ |
| | $z = 0$ | $95.14 \pm 0.20$ | $0.34 \pm 0.02$ | $\mathbf{95.34 \pm 0.14}$ | $\mathbf{0.20 \pm 0.01}$ |
| | $z = 0.2$ | $\mathbf{94.06 \pm 0.27}$ | $0.42 \pm 0.04$ | $93.80 \pm 0.26$ | $\mathbf{0.31 \pm 0.03}$ |
| Fashion-MNIST | $z = -0.2$ | $75.80 \pm 1.01$ | $1.09 \pm 0.05$ | $\mathbf{80.04 \pm 0.70}$ | $\mathbf{0.83 \pm 0.03}$ |
| | $z = 0$ | $79.42 \pm 0.77$ | $1.13 \pm 0.04$ | $\mathbf{81.12 \pm 0.49}$ | $\mathbf{0.79 \pm 0.03}$ |
| | $z = 0.2$ | $\mathbf{79.72 \pm 0.78}$ | $1.45 \pm 0.11$ | $78.55 \pm 0.82$ | $\mathbf{0.92 \pm 0.04}$ |

The comparison results of the experiment are highlighted in bold font.

Parameter $\varepsilon$. We established various privacy demand intervals in the following tests to assess the viability of PDP-SGD, because the PDP algorithm depends on the choice of privacy budget intervals. Specifically, beginning from 0.6 to 2, the maximum value of the privacy interval is increased by 0.2 increments, while the lowest value is maintained at 0.5. A wider privacy interval denotes a laxer necessity for privacy. In trials using the MNIST dataset, we set $z = -0.2$, while in studies using the Fashion-MNIST dataset, $z = 0.2$.

In several sets of trials with various privacy intervals, the proposed PDP-SGD delivers flat baseline or better-than-baseline accuracy, as shown in Figure 6. There are fewer training rounds for shorter intervals since the privacy budget interval affects the overall number of

training rounds in the trials. It is clear that PDP-SGD performs worse than the baseline with fewer training rounds—the method of restricting the model parameters by dissimilarity necessitates more rounds of updating to prevent entering a local optimum solution.

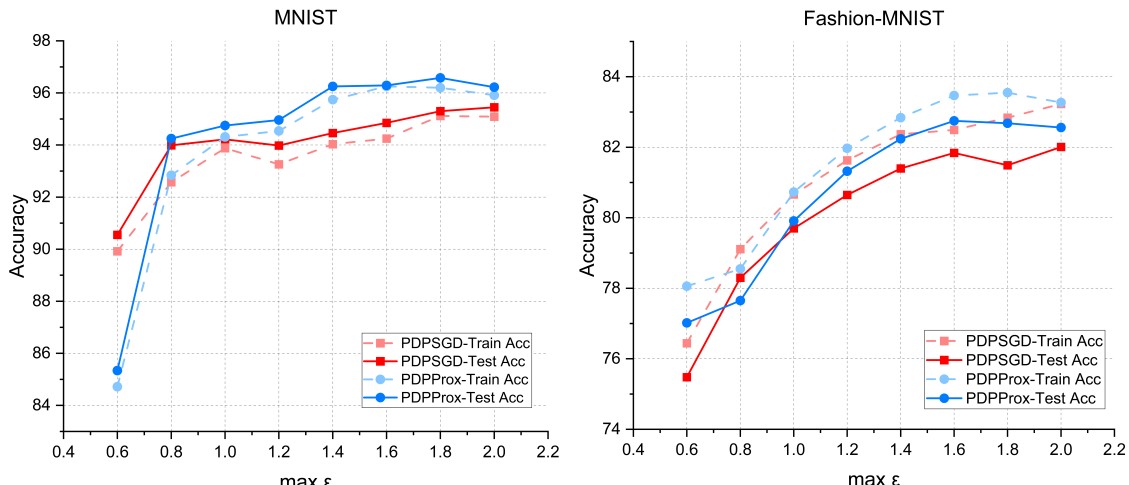

**Figure 6.** Impact of parameter $\epsilon$ on the accuracy of model testing.

## 6. Conclusions

This paper investigates how well FedProx is optimized for currently used algorithms in different scenarios involving privacy preservation. In this study, we merge the concepts of DP and PDP with the FedProx framework and propose two classes of algorithms, DP-Prox and PDP-Prox, to comprehensively evaluate the utility of FedProx for privacy algorithms to address heterogeneity and improve model accuracy. The experimental results show that the proposed algorithms converge significantly better than the baseline algorithm under the heterogeneity condition and the convergence performance is more stable; through full validation under the PDP setting, PDP-Prox can obtain higher model accuracy.

In our current research work in combining FedProx and DP, there are shortcomings in various aspects such as the setting of imprecise values and the effect of system heterogeneity on the recycling of privacy budgets. As a future direction of work, we would like to standardize how to base the selection of imprecise values on practical situations in experiments on heterogeneity; in terms of privacy protection, on the other hand, we would like to carefully categorize and select user privacy levels and further investigate the allocation and utilization of privacy budgets under conditions of system heterogeneity.

**Author Contributions:** Conceptualization, T.A. and L.M.; methodology, L.M.; software, L.M. and Y.Y.; validation, L.M., Y.Y., J.W. and Y.C.; formal analysis, L.M., J.W. and Y.C.; investigation, L.M.; resources, T.A. and W.W.; data curation, J.W.; writing—original draft preparation, L.M. and Y.Y.; writing—review and editing, T.A. and W.W.; visualization, L.M.; supervision, T.A. and W.W.; project administration, T.A. and W.W.; funding acquisition, T.A. All authors have read and agreed to the published version of the manuscript.

**Funding:** This research received no external funding.

**Data Availability Statement:** No new datasets were generated in this paper. The synthetic dataset used in the experiments can be accessed at https://github.com/litian96/FedProx on 13 September 2019; both real datasets used in the experiments can be accessed through the API provided on the official tensorflow website.

**Acknowledgments:** We are very grateful to the university and college authorities for their support and guidance, and to our research partners for their contributions. We are very thankful to our supervisors for their guidance and assistance.

**Conflicts of Interest:** The authors declare no conflict of interest.

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
