# Peer review of "Consideration of FedProx in Privacy Protection"

_electronics, doi:10.3390/electronics12204364_

Round 1
Reviewer 1 Report
The paper proposes an algorithm that combines FedProx with differential privacy and personalized differential privacy. The algorithm is well described, and the results, which are mostly well presented, support its validity. The manuscript is missing a motivation, which warrants me to give a minor revision recommendation. However, extensive misuse of citations forced me to opt for major revisions: any inappropriate self-citations must be removed before I can give a positive overall recommendation on the research.
Major Revisions:
- Include a motivation for this study, and what are the real-world advantages of using DP/PDP-Prox over alternatives;
- Discuss the limitations of the proposed solution, possibly in the conclusion which feels quite short;
- Completely revise the citations. Mentioning "incentive mechanisms" and attaching 5 citations, many of which are unrelated self-citations, is not acceptable. First of all, I like to see citations mentioned one at a time: for example, "[2]" is good practice, "[2,3]" should only be used if the two works are very similar (e.g., [3] is a strictly incremental work to [2]), "[2-6]" should never be used. Additionally, and perhaps most importantly, do not cite unrelated works and especially do not inappropriately self-cite. I believe that citations number 2, 3, 4, 5, and 6 are all inappropriate and should be removed.
Minor Revisions:
- It's "Federated Learning", not "Federal Learning"
- Highlighting results in tables is good practice, and you did it. It is common to highlight results in bold, not using red or any other color. And for consistency, if the best results are highlighted, these should be highlighted even when the best results are from another method.
- While included in fairly high res, figures 1, 2, and 3 are obviously images. Figures 4 and 5 seem ok, but nonetheless no image has searchable text: I would suggest to include those graphs as PDFs instead, so they are accessible and indexable.
Reviewer 2 Report
My main issue with this paper is regarding its novelty. What exactly is the contribution here? To me, authors just combine two existing techniques, i.e., FedProx and DP, and show that this outperforms the case with FedAvg + DP in heterogeneous environments. This is sort of obvious as FedProx is specifically designed to combat heterogeneity. What is new here? What is the main technical challenge and outcome of the paper?
The experimental evaluation is limited to only MNIST and Fashion MNIST datasets, which are significantly trivial datasets. Experiment details are also lacking. For example, in section 5.2, authors say they adjusted the system heterogeneity to 0, 50%, 90%. What does this mean? What do you adjust? How do you adjust? Where is the case with dataset heterogeneity?
1. The writing of the paper needs serious improving. Many of the sentences feel automated. Better flow is needed to convey the main messages and results of the paper.
2. The paper includes many spacing errors and typos. For example the use of "agression" instead of "aggregation" or spacing error on page 3 line 110. Another one is the missing "min" on the right hand side of equation 7. Please proofread the entire manuscript and fix the typos and grammatical errors.
In its current form, it is not ready for publication.
Round 2
Reviewer 1 Report
All of my comments have been properly addressed, so I am giving an accept to this revised manuscript.
Reviewer 2 Report
I still think that the contributions of this manuscript are limited and the fact that only MNIST and Fashion MNIST datasets (aside from the synthetic ones) are used in experiments is a contributing factor to this limitation.
However, the manuscript has improved compared to the last version.
The language of the paper has improved but there is still room for improvement. For example, there is no space between "system heterogeneity" and "statistical heterogeneity" subheadings in Section 5.2. I am sure there are others too. So, I suggest authors to do another proofreading of the manuscript.